# Effect of Herbal Addition on the Microbiological, Oxidative Stability and Sensory Quality of Minced Poultry Meat

**DOI:** 10.3390/foods10071537

**Published:** 2021-07-02

**Authors:** Danuta Jaworska, Elżbieta Rosiak, Eliza Kostyra, Katarzyna Jaszczyk, Monika Wroniszewska, Wiesław Przybylski

**Affiliations:** 1Department of Food Gastronomy and Food Hygiene, Institute of Human Nutrition Sciences, Warsaw University of Life Sciences-SGGW, Nowoursynowska Str. 166, 02-787 Warsaw, Poland; elzbieta_rosiak@sggw.edu.pl (E.R.); jaszczyk.katarzyna@wp.pl (K.J.); wroniszewska.monika@gmail.com (M.W.); wieslaw_przybylski@sggw.edu.pl (W.P.); 2Department of Functional and Organic Food, Institute of Human Nutrition Sciences, Warsaw University of Life Sciences-SGGW, Nowoursynowska Str. 166, 02-787 Warsaw, Poland; eliza_kostyra@sggw.edu.pl

**Keywords:** meat product, dry herbs, herbal extracts, shelf life, sensory characteristics, microbial quality

## Abstract

The study aimed to assess the effect of herbal additions with antioxidant properties (pepper, thyme and oregano) on the microbiological and oxidative stability as well as the sensory quality of minced poultry meat. Meatballs treatments without additives and treatments with the addition of three types of spices in two forms—dry spices and industrial extracts were examined. Popular seasoning additives of oregano (*Origanum vulgare*), thyme (*Thymus vulgaris*) and black pepper (*Piper nigrum*) at 0.3% of dry herbal or 0.003% as industrial extract were added to minced meat. The microbiological, chemical and sensory tests were performed at specified times and storage temperature. Based on the studied criteria, products maintained constant and adequate quality by up to 10 days while stored in 4 °C. In the case of all tested samples, the overall sensory quality began to deteriorate after 10 days of storage. The preservative role of herbs and extracts in meat products during processing and storage was observed. Oregano and black pepper in both forms maintained good microbial quality and showed their inhibitory effects on the growth of psychrotrophic bacteria. It was observed that dried herbs revealed a stronger antioxidant effect than additives in the form of extracts. The studied dried herbs played an antioxidant, antimicrobial and preservative role in meat products during processing and storage.

## 1. Introduction

The consumption of meat and meat products has increased significantly in recent years, especially poultry meat is eagerly chosen by consumers. This is partly due to its relatively low price compared to other meat types. Not all meat is utilized fresh; unfortunately, prolonged storage of meat and meat products results in adverse changes affecting their quality and causes quality deterioration [1]. The most commonly described changes occurring in meat products during storage are lipid oxidation [2,3] and microbiological transformations [4]. These processes significantly reduce the storage stability of products. The dynamics of these processes can be affected by many factors, including the chemical composition of meat (water content and fatty acid composition), access to light and temperature.

To prevent lipid oxidation processes, synthetic antioxidants are often used, but their health effects are questionable due to potential human health risks and increased toxicity of the product [5]. Many studies confirm the negative impact of excessive consumption of synthetic food additives on human health resulting in strict regulations of their use in food products [6,7]. Nevertheless, they are still used due to their low price, high stabilizing effect and high effectiveness [8].

Therefore, natural antioxidants could be an alternative solution for this problem since plants and plant materials are rich in bioactive compounds (e.g., natural antioxidants) with potential beneficial health effects. Herbs and spices have been used for centuries to prepare meals. They are used in many dishes to give the palatable taste and aroma or extend the shelf life of food products [9]. Moreover, as the interest of consumers in using natural products is increasing, there is a need to study alternative antioxidant sources [6,10].

To prolong food shelf life many different processes and additives are applied, among them spices and herbs [11]. The chemical nature of natural antioxidants has the advantage of being more acceptable by consumers since they are considered less hazardous to health [12,13]. Considering their natural origins, bioactive compounds obtained from plants (black pepper, oregano) are ideal candidates to replace synthetic antioxidants (generally considered as less safe) to increase the shelf life of meat products [14]. The antioxidants can also improve the stability of lipid-containing foods, preventing loss of sensory and nutritional quality [15].

The raw material used in various branches of the food industry is thyme herb which gives flavour and aroma to prepared dishes (meat, sausages) [16]. Thyme is an aromatic plant used as a spice and for medicinal reasons. It also provides aroma in food, has preservative, antioxidant, antifungal and antimicrobial activity [16]. In another popular spice, oregano, essential oils like thymol and carvacrol are present. In addition, it contains active compounds such as tannins, flavonoids—including apigenin and luteolin, sesquiterpenes, organic acids, phenolic acids, sluices, resin compounds, phytosterols, vitamin C and mineral compounds [17,18].

Pepper fruits and the oil obtained from them are characterized by having antiseptic, antibacterial, strengthening the immune system or anti-inflammatory properties [19]. Black pepper, due to its antioxidant properties is used in various meats [20]. However, the routine application of plant extracts, herbs, spices and essential oils as an antioxidant in the food industry is still uncommon mainly because of limited data about their effects in different meat products [7].

All products introduced to the marketplace have to gain consumers’ interest and maintain their sensory characteristics through their normal shelf life. This can be verified by evaluating the products by sensory trained panels. Therefore, the study aimed to assess the effect of herbal additives with antioxidant properties (pepper, thyme and oregano) added in the dry form and industrial extracts on the microbiological and oxidative stability and sensory quality of minced poultry meat.

## 2. Materials and Methods

### 2.1. Materials and Reagents

The poultry breast meat was delivered directly from a commercial meat processing plant located in Wielkopolska (Poland). Herbal additives (dry spices) and pork jowl were purchased in the retail store whereas industrial extracts—oleoresins produced by the German company Rüther Gewürze Gmbh were purchased from a Polish distributor.

All reagents were of analytical grade. For TBARS a solution of 2-thiobarbituric acid (TBA) and trichloric acid (TCA) was delivered by SigmaAldrich, Germany. Buffers for calibrating pH meter were obtained from WTW company (Weilheim, Germany).

Reagents for the total viable count (TVC) of mesophilic bacteria, psychrotrophic bacteria were coming from: Nutrient Agar AT 2% from Biokar Diagnostics Warsaw, Poland; Ref. No. BK185HA; for *Bacillus* spp. and *Clostridium* spp. the genus were determined respectively using Mannitol Egg Yolk Agar supplemented with Polymyxin—MYP Agar (Bio-Rad, Watford, UK, Ref. No. 3569604) and Sulfite Polymyxin Sulfadiazine—SPS Agar (Merck, Darmstadt, Germany, Ref. No. 110235). The sterile plastic filter bags were delivered by Interscience, Saint Nom la Brétèche, France). Sterile buffered peptone water was obtained form LabM, A, Neogen Co., Lansing, MI, USA. Plates bags with oxygen absorbers for incubation Clostridium spp. were delivered by Generator Genbag anaer, Biomerieux, ref. No 45534.

### 2.2. Manufacture of Minced Poultry Meat

The experimental material consisted of poultry meatballs. The formulation of the experimental material including levels of herbal additives was determined during the preliminary research (based on scientific literature and sensory tests). The final composition of the experimental poultry meatballs is presented in Table 1.

A control sample was prepared without herbal additives and the experimental material was meatballs with the addition of three different types of spices in two forms—dry spices and industrial extracts (oleoresins). The popular seasoning additives oregano (*Origanum vulgare*), thyme (*Thymus vulgaris*) and black pepper (*Piper nigrum*) were used at levels listed in Table 1.

The meat raw material (poultry and pork jowls) was ground through a Zelmer grinder (model ZMM, 1900 W, Rzeszów, Poland) with 8 mm diameters plates. All ingredients were mixed thoroughly, then balls with a diameter of 3 cm were hand-formed. The heat treatment process was carried out using steam in a convection-steam oven (RedFox KE-423, RM Gastro, Veselí nad Lužnicí, Czech Republic) for 10 min at 200 °C with 30% steam. The temperature was controlled by a thermocouple (82 °C in the geometrical center of the experimental material). After cooling to the temperature of about 21 °C, the meatballs were placed in thermostable polyethylene-polyamide pouches (Hendi, Gądki, Poland) and vacuum packed using a vacuum packaging machine (No 691310, Stalgast, Warsaw, Poland) with a pressure of 9 bars. The vacuum time was 30 s and the sealing time was 9 s. The packed products were refrigerated at 4 °C ± 1 °C. All measurements were carried out on days 0, 5, 10, 15, 18 and 21 of the storage period.

### 2.3. Physical-Chemical Methods

#### 2.3.1. pH Value

The pH value was measured using a pH meter pH 330i/Set, (WTW—Wissenschaftlich-Technische Werkstatte, Weilheim, Germany). Before performing the determinations, the device was calibrated according to buffers at pH 4 and 7. Measurements were made on the days specified in the research schedule storage day and were made in each of the three meatballs taken randomly from each research group. All measurements were carried out on days 0, 5, 10, 15, 18 and 21 of the storage period. The product temperature during the measurement was equal to the ambient temperature and was 21 ± 1°C.

#### 2.3.2. Water Activity (aw)

Water activity was measured using AquaLab 4TE (Aqua Lab, Decagon Devices, Pullman, Washington, DC, USA) apparatus with an accuracy of 0.001 at a temperature of 21.0 ± 1.0 °C. A measuring cell was filled with an individually prepared sample. Measurements were made on the days specified in the research schedule storage day in each of the three meatballs taken randomly from each research group. All measurements were carried out on days 0, 5, 10, 15, 18 and 21 of the storage period.

#### 2.3.3. Thiobarbituric Acid (TBARS) Method

The content of malonic aldehyde as an indicator of the degree of lipid oxidation was determined by the method of Shahidi [21] and expressed as the number of TBARS in mg of malonaldehyde (MDA) per 1 kg of the tested sample. Aldehydes present in fat under the influence of high temperature formed colorful complexes with. The color intensity of the solution was measured on a Hitachi U-1100 spectrophotometer (Massachusetts, United States). Absorbance measurements were made on the days specified in the research schedule storage day in each of the three meatballs taken randomly from each research group at 532 nm. All measurements were carried out on days 0, 5, 10, 15, 18 and 21 of the storage period.

### 2.4. Microbiological Methods

Microbiological analysis was performed using the surface plate culture technique. In poultry meatball stored under anaerobic conditions. Most Clostridia are sulfite reducers, among them *C. perfringens* and *C. botulinum* are the species most frequently involved in food poisoning.

The microbial suspension or decimal dilutions (1 mL) were spread on solidified appropriate medium in duplicate. The tested material was taken with a sterile pipette and spread with a sterile pad on the surface of the medium until absorbed and incubated at the appropriate temperature depended on the analysis (TVC. *Bacillus* spp., *Clostridium* spp.—37 °C; psychrotrophic—7 °C for a specified time (24–48 h up to 10 days in the case of psychrotrophic bacteria).

Microbiological analysis of each meatball was made immediately after production and then on the days specified in the research schedule. From the poultry meat, 10 g samples were weighed in sterile conditions and placed in plastic filter bags and 90 mL of 0.1% *w*/*v* sterile buffered peptone water was added and then the mixture was homogenized for 1.5 min in homogenizer (Masticator 80, IUL) with normal speed. One ml of the prepared sample or progressive 10 fold dilutions in sterile peptone water (0.1% *w*/*v*) were used. The analysis was based on TVC bacteria ISO standard [22], psychrotrophic bacteria were counted after 10 days of incubation in 7 °C. Enumeration of *Bacillus cereus* was performed in accordance with ISO standard [23].

*Clostridium* spp. plates were incubated in bags with oxygen absorbers. To active spores of the *Clostridium* spp. genus, the material was incubated in a 75 °C water bath for 10 min before plating.

The growth of black colonies indicating *Clostridium* spp. bacteria on SPS Agar was not observed, cremo-white colonies were counted instead. The yellow colonies were counted on MYP Agar for the enumeration of *Bacillus* spp. *Bacillus cereus* grows as pink colonies are often surrounded by a zone of precipitate indicating the production of lecithinase. All growing colonies were counted for TVC and psychrotrophic analysis.

### 2.5. Sensory Method

To determine the sensory quality of the samples, an analytical method—QDP (Quantitative Descriptive Profile) was performed, based on ISO standard [24]. The basic assumption of this method is that aroma, flavour and texture are not individual quality features but are considered as a complex of many individual attributes. They can be distinguished, identified and their intensity determined. Seventeen defined sensory attributes were measured to quantify the quality of the tested samples: 5 odour attributes (meaty, sour, fatty, herbal and other), 3 attributes of appearance (surface, homogeneity, cross-section), 2 texture attributes—juiciness and softness—and 7 attributes of taste/flavour (meaty, sour, fatty, salty, bitter, herbal, other). Based on all the above-mentioned attribute characteristics, a sensory trained panel assessed an overall sensory quality (very low–very high) for each sample (Table 2). An unstructured, linear scale of 100 mm converted to numerical values (0–10 conventional units, c.u.) was used. The anchor marks of the tested attributes are presented in the Table 2. The average result was based on a minimum of 18 individual results.

The trained panel consisted of 9 members (7 women and 2 men, age 28–58), who were extensively and formally tested before being selected, according to the ISO standard [25]. The panellists had 4 to 18 years of theoretical and practical experience with sensory procedures and evaluation of different food products with various methods (including profiling). The assessors’ ability to differentiate product samples by various concentrations of volatile and non-volatile stimuli was verified. They underwent descriptive tests using a series of food products in which they described the sensory characteristics of the samples. Between the subsequent evaluations, the assessors were provided water to neutralize the taste/flavors of the samples.

#### 2.5.1. Sample Preparation

Meatballs weighing about 10 g each were cut in half, yielding two parts with a diameter of about 3 cm. The prepared meatballs were placed in disposable plastic containers and covered with a lid and then coded with a three-digit code. They were served 4 samples at the same time in random order. This rule limits the impact of the carryover effect (i.e., the impact of a previous sample on a subsequent sample).

#### 2.5.2. Evaluation Conditions

Sets of samples for sensory evaluation were presented at an ambient temperature of 21 ± 1 °C and evaluated within two hours. While carrying out sensory assessments constant temperature, lighting and elimination of distracting factors like noise and off odours were applied. All measurements were carried out on days 0, 5, 10, 15, 18 and 21 of the storage period.

### 2.6. Statistical Methods

The statistics 13.3 PL (Microsoft, Redmond, WA, USA) program was used to compare the results. The normality of distribution of all analyzed traits was checked using the Shapiro–Wilk test. To determine the difference between the samples, a one-way ANOVA analysis of variance was used. The significance of differences between individual means was determined using Tukey’s post-hoc test (RIR). The α value 0.05 was used. Additionally, multiple factor analysis (MFA) for attributes of sensory analysis was applied using Statistica 13 version software (TIBCO Software Inc., Palo Alto, CA, USA, 2017). The aim of using multidimensional statistical analysis MFA was to assess the contribution and strength of the impact of individual attributes in creating and explaining the general variability of the sensory quality of the studied material.

## 3. Results

### 3.1. pH and Water Activity Values

Average results regarding changes in pH value are presented in Table 3. Analysis of the obtained results shows that the pH value did not change considerably during the established time intervals. The pH value varied from 5.96 up to 6.36 depending on the sample and time of storage.

Analysis of the water activity (a_w_) in the samples (Table 4) indicates that the tested material contained a high amount of water, which does not guarantee storage stability and long shelf life. Analysis of Table 4 indicates that during the observed period, a slight, non-significant increase of a_w_ was observed.

### 3.2. Lipid Oxidation

Figure 1 shows the effect of spice additive, both in the form of oleoresins and in the dried form, on the processes of fat autoxidation. In each sample with the spice additive, both in the form of oleoresins and in the dried form, the processes of fat autoxidation were slowed.

I was found that dried herbs had a stronger antioxidant effect than the additives in the form of oleoresins. The level of lipid oxidation transformation increased significantly with 21 storage time and the content of malonaldehyde—MDA was highest in control samples comparing to others.

### 3.3. Microbiological Quality

#### 3.3.1. Assessment of TVC Quality of Meat Products with the addition of Spices in Dry and Extracts Form

The TVC of control samples was approximately 0.65 log cfu/g initially but increased with storage time, reaching close to 3.46 log cfu/g at 21 days (Figure 2). The oregano (extract and dry form) did not affect product microflora in the stored period until the last day when a significant statistical (*p* < 0.05) inhibitory effect was observed. At the beginning of the experiment, significantly higher microbiological contamination of the product samples was found with the addition of thyme in the form of extracts. During storage, the phenomenon that was originally presented was negligible. By the end of the storage period, TVC values in and pepper dry form-treated samples were significantly higher compared to the control sample and exceeded the amount of 4 log cfu/g contamination. Regarding the oregano, the dry form-treated sample TVC value was significantly lower compared to the control sample. On the last day of the analyses, significant differences in the number of TVC were found in the samples with the addition of dry spices and the extracts form.

#### 3.3.2. Assessment of Psychrotrophic Bacteria Quality of Meat Products with the Addition of Spices in Dry and Extracts Form

In the case of psychrotrophic bacteria (Figure 3), no bacterial growth was found in the treatment or control samples until the 15th day of refrigerated storage. On the last day of refrigerated storage, significantly lower (*p* < 0.05) contamination values were found in the case of oregano and black pepper (both form), compared with the control sample which indicates the inhibitory effect of spices on the number of psychrotrophic bacteria. However, no difference was noted between the control sample (4.97 log cfu/g) and the samples with thyme in a dry (4.69 log cfu/g) or extracts (5.14 log cfu/g) form.

#### 3.3.3. Assessment of Bacillus spp. and Clostridium spp. Bacteria Quality of Meat Products with the Addition of Spices in Dry and Extracts Form

The bacteria of genus *Bacillus* spp. (Table 5) was found in the control sample on the tenth day of refrigerated storage and the number grew steadily until the end of the storage period when it reached value 2.43 log cfu/g. In the case of the addition of oregano in the oleoresin form, significantly higher numbers of *Bacillus* spp. bacteria on the fifteenth and eighteenth days of storage were found compared to the control sample. *B. cereus* was found in the product with the addition of thyme in the form of oleoresin on the 10th and 15th day of storage. Table 5 presents also the results obtained for total anaerobe microorganisms in tested products only with the addition of herbal extracts.

### 3.4. Sensory Quality of Meat Products with the Addition of Spices in Dry and Oleoresin Form

In Figure 4 changes in overall sensory quality attribute during 21 days of storage are displayed. For clear presentation, samples in which no sensory changes were observed were omitted. During 10 days of storage, no significant changes regarding overall sensory quality were observed in the case of herbal additives and industrial extracts. It is showed that this descriptor significantly decreased intensity level. When it comes to all the tested samples the overall sensory quality began to deteriorate after 10 days of storage. It must be pointed that of the control sample the process was most intensified. Additionally, the dry herbal additives were more effective in quality preservation compared to industrial extracts, in which the decrease of overall sensory quality was greater. Only black pepper extract gave a similar quality level in comparison to black pepper powder.

The data collected in Table 6 indicates that the first two principal components based on QDP data explain 69% of the original variability of overall sensory quality (dry herbal additives) and 66% of total variability concerns products containing industrial extracts. The strongest negative influence on the overall sensory quality of the tested material was observed when dry herbal additives were applied, it was accompanied by the intensity attributes such as sour odour and flavour and bitter taste, as well as other odour and flavour (named rancid). Similar observations were stated for meatballs with industrial extracts (Table 6).

## 4. Discussion

The observed values for pH are typical of these types of meat products. Similar values of pH in poultry meat ranging from 5.81 up to 6.23 were observed in other studies [13,26].

Herbal additives or industrial extract addition did not have any significant effect on water activity in this study. Such levels of this parameter are typical and were observed in other studies [27]. The relatively high values of water activity indicated that the growth rate of microorganisms in the tested samples was not limited and the treatment product had low stability. Water activity is an important parameter concerning the state of water in food. It determines to a large extent microbiological, chemical and biochemical stability, as well as physical properties of food products. Optimal water activity for product shelf life is similar to the water content corresponding to the minimization of the rate of all processes that lead to spoilage of food, whereas a_w_ values below 0.6 indicate the microbiological stability of the product.

In each test with the spice additive, both in the form of oleoresins and in the dried form, the processes of fat autoxidation were slowed somewhat. Similar research results were obtained by Fasseas et al. [28]. In their research, the effect of 3% addition of oregano and sage oils on the quality of raw and heat-treated meat stored in refrigeration conditions was estimated. They showed that the addition of oils inhibits the auto-oxidation processes of fats and oregano oil proved to be a more effective antioxidant. Oregano has antibiotic and antioxidant effects, along with other biological activities. Moreover, they noticed a statistically significant increase in the TBA index concerning the longer storage time of a control sample. In the study conducted by Gramza-Michałowska et al. [29], thyme addition and rosemary extract inhibited lipid oxidation. The results presented in this study showed a more significant effect after using dry herbs. In another study [30], the effect of rosemary and thyme extracts (0.05%) on lipid stability and the protein nutritional value of frozen-stored fried balls from ground pork was investigated. It was found that rosemary and thyme limited lipid oxidation, as well as reduced changes in methionine and lysine content and protein digestibility were comparable to the control sample. Similarly, in the study of Sariçoban et al. (2014) it was found that the addition of thyme essential oils decreased oxidation and microbial growth, extending the shelf life of chicken pâté at 0.05% [31]. The results of that study indicated that the addition of herbal extracts to turkey meatballs was effective in controlling lipid oxidation and microbial growth, during the refrigerated storage of pre-cooked turkey meatballs [32]. Manhani et al. [33] compared the antioxidant potential of rosemary and oregano deodorized, commercial extracts (0.04% addition) in precooked beef hamburgers by assessing the changes in lipid oxidation. A lower concentration of TBARS values was observed in this research. However, the obtained results showed that the applied dry herbs and essential oil addition to poultry meatballs were not sufficient to reduce the radical formation and decrease the lipid peroxidation. The amounts used did not prevent the deterioration of sarcoplasmic proteins and decrease deterioration reactions and therefore cannot extend the shelf-life of the studied product.

The oregano (extract and dry form) did not affect product TVC count of microflora until the last day of cold storage of sample when a significant (*p* < 0.05) inhibitory effect was observed. However, Hernández et al. [34] found that the application of oregano essential oil in dried meat was effective in inhibiting *Salmonella enteritidis* and *Escherichia coli*. In the case of thyme, both dry and as the extract, the microbiological effect was not significant. The pepper dry-treated samples were significantly (p < 0.05) higher contaminated compared to the control sample and exceeded the amount of 4 log cfu/g, contamination of TVC count. Similar results were obtained in the study of Grabowska et al. [35] concerning two thyme samples, while the microbiological quality of 4 others samples was classified as acceptable. According to Grabowska et al. [35], the microbiological quality of black pepper depends on the pre-treatment disinfection, e.g., washing in hot water, which results in a significant impact on the total number of mesophilic aerobic microorganism reduction.

The results obtained by other authors indicate that thyme extracts form showed the weakest antimicrobial effect compared to oregano and garlic [36]. The observed last day of cold storage effect can be considered as the inhibitory effect of the spices added in the form of oleoresins. The spices’ extracts used as an ingredient of poultry meatball did not increase, in any of the analysed samples, the TVC count of mesophilic microorganisms. Microbiological contamination of spice demonstrated based on 425 notifications to the RASFF system in the period 2004–2014 showed the contamination up to 2.5 × 10^5^ cfu/g [37].

No difference of psychrotrophic bacteria was noted between the control sample (4.97 log cfu/g) and the samples with thyme in a dry (4.69 log cfu/g) and extracts (5.14 log cfu/g) form. Similar results were obtained by Agrimoni et al. [38] studying the effect of essential oils in the form of an emulsion applied to cellulose pads on the psychrophilic microflora of ground beef. *P. fluorescens* constituting 90% of the saprophytic microflora of minced meat was the most sensitive bacterium to emulsions composed of oregano, thyme and cumin. The researchers reached a shelf life of 12–15 days. Moreover, in Talebi et al. [39] study higher concentrations (1% *v*/*v*) of essential oils of Mentha piperita and Bunium persicum extended the ground beef shelf life from 4 to 7 days. The obtained of own results research in case of spices added in dry and extracts form of oregano and black pepper of long-stored refrigerated meat products are a good forecast for the technology of ready-to-eat products since the addition of spices effectively inhibited the group of microorganisms dominant in these products. The weakest effect was found in the case of thyme, which could be more effective when combined with other spices.

The number of *Bacillus* spp. bacteria determined in the analysed products constituted the lower limit of *Bacillus* spp. contamination showed by RASFF reports from 2004–2014 (1 × 10^3^–3 × 10^8^ cfu/g) for spices, which indicates the good microbiological quality of the spices used [37].

The addition of oregano in the dry form had a significant inhibitory effect compared to the control sample and the sample with the addition of oregano in the form of an oleoresin. In the case of products with the thyme added in the form of oleoresin the analysis carried out in the storage periods of 18 and 21 days showed the insignificant decrease in the number of *Bacilllus* spp. bacteria compared to control and dry added sample. These results are opposite to described in the literature hierarchy of antibacterial activity of oleoresins (essential oils), which shows thyme as the least antimicrobial effective: *Allium sativum > Origanum vulgare > Thymus vilgaris > Ocimum basilicum* [36]. The addition of black pepper (dry and oleoresin forms) did not have a significant effect on the number of *Bacillus* spp. bacteria in the analysed products. Reports from RASFF and European Scientific Agencies Report show that *Salmonella* spp. and pathogenic *Bacillus* species are the most common hazards as for black pepper and other herbs [37]. These own results confirmed good microbiological quality of the herbs used in the experiments, which is not typical of given seasoning and is subjected to fluctuations depending on various factors like technological process and transport [35]. The only exception was B. cereus determined in products with the addition of oleoresin thyme on the 10th and 15th day of storage. The sulphite reducing anaerobe bacteria that form black colonies due to iron sulphide sediment were not determined in products treated with the oleoresins spice extracts the added spices (Table 5). The RASFF report showed sulphite reducing anaerobe bacterial contamination at the level of 2 × 10^3^ cfu/g in less than 1% of the samples [37]. Only oregano revealed an inhibitory effect on the anaerobe microflora of meat products after 18 days of cold storage. Krishnan et al. [40] found a significantly lower count of *Lactic Acid Bacteria* (LAB) and *B. thermosphacta,* both of which are facultative anaerobic species of the meat products, in the samples treated with extracts of clove, cinnamon and oregano.

Referring to the sensory part of the research, Manhani et al. [33] compared oregano commercial extract (0.04% addition) in precooked beef hamburgers by assessing the changes in sensory quality (colour, taste, odour evaluation) during 30 days of frozen storage. They showed that at the end of storage time the hamburgers with oregano extracts were characterized by satisfactory sensory quality compared to the control sample. Moreover, in the case of meatballs with extracts, the fatty odour and flavour attributes played a significantly negative role. As it was expected, the meaty odour and flavour as well as herbal odour and flavour were of positive correlation and were responsible for increasing the overall sensory quality of the studied poultry products. Spicy flavour and aromas usually do not come from a single compound but result from the presence of various compounds, including terpenes, essential oils and aldehydes in plant products and foods. A lot of these substances are volatile, which gives them a potent fragrance. Therefore, they have a great influence on creating the aroma and flavour of products [41].

In the presented study it was shown that some attributes, especially those with a negative sensory influence, determined the overall sensory quality of the examined products and were of crucial importance for the overall sensory characteristics. After exceeding a critical value, negative attributes strongly decreased the overall sensory quality of the tested products. It is known from the literature that even a slight increase in the intensity of negative attributes, mainly flavor and especially odor, is connected with a severe decrease of the overall sensory quality of food products. These relations are not linear [42].

The sensory quality of foods is one of the main factors influencing the acceptance of a product on the market. For this reason, studying factors that influence sensory quality is crucial in food product development. The study shows that herbs and extracts of spices allow creating new meat products of high sensory quality. However, the high intensity of essential oils and spicy flavour of spice extracts might be a limitation in the application of the extracts. For this reason, masking negative attributes like bitterness can be a crucial task in the development of new products with spice addition. On the other hand, the results of the studies indicated [41] show that, obviously, not all consumers are averse to bitter tastes.

## 5. Conclusions

The studied dry herbs play an antioxidant, antimicrobial and preservative role in poultry meatball type products during processing and storage. These compounds have shown the potential to inhibit or delay oxidation. The shelf-life and the microbiological quality and safety of the meat products stored at a low temperature of 4 °C can be enhanced by the addition of oregano and black pepper in dry or extracts form. Spices should be composed as a mixture due to the different effects of spices on the microflora of cooled meat products. Based on sensory criteria, 10 days of storage provided constant and adequate quality of steamed poultry products stored at low temperature. It was observed that dried herbs showed a stronger antioxidant effect than extracts. The correlation between the economics of antioxidant use and the economics of oxidation spoilage should also be considered before drawing any conclusions for the meat industry.

## Figures and Tables

**Figure 1 foods-10-01537-f001:**
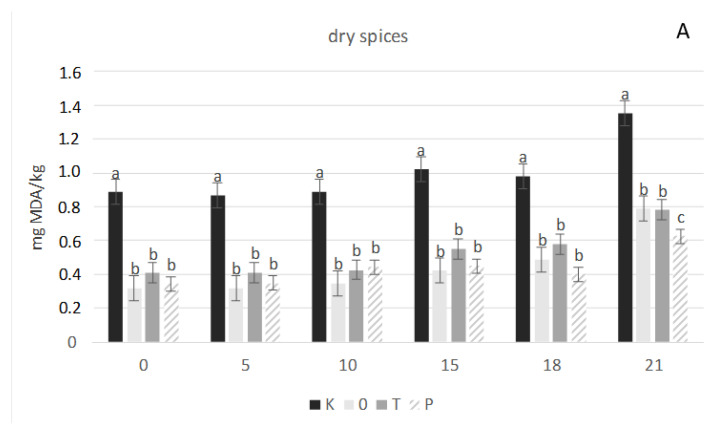
Results of TBARS measurements carried on products with the addition (**A**) of herbs in dried form and (**B**) in the form of oleoresins on the day of preparation and during 21 days of storage at temperature 4 ± 1 °C, days of storage: 0; 5; 10; 15; 18; 21. K—sample control, O—oregano, P—black pepper, T—thyme, OO—oleoresin oregano, OP—oleoresin black pepper, OT—oleoresin thyme, MDA—malonaldehyde; a, b, c—means with different superscripts differ significantly between various products on the day of storage (*p* < 0.05).

**Figure 2 foods-10-01537-f002:**
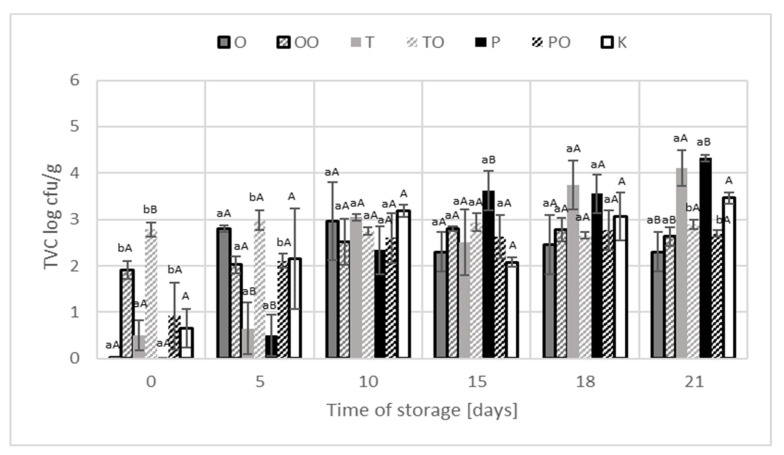
Total viable count of mesophilic microorganisms in examined products and control sample during storage at temperature 4 ± 1 °C. ^ab^ statistic differences (*p* < 0.05) in bacteria count between products with dry and oleoresin spices. ^AB^ statistic differences (*p* < 0.05) in bacteria count between products with spices and control sample. K—sample control, O—oregano, P—black pepper, T—thyme, OO—oleoresin oregano, OP—oleoresin black pepper, OT—oleoresin thyme, TVC—Total Viable Count.

**Figure 3 foods-10-01537-f003:**
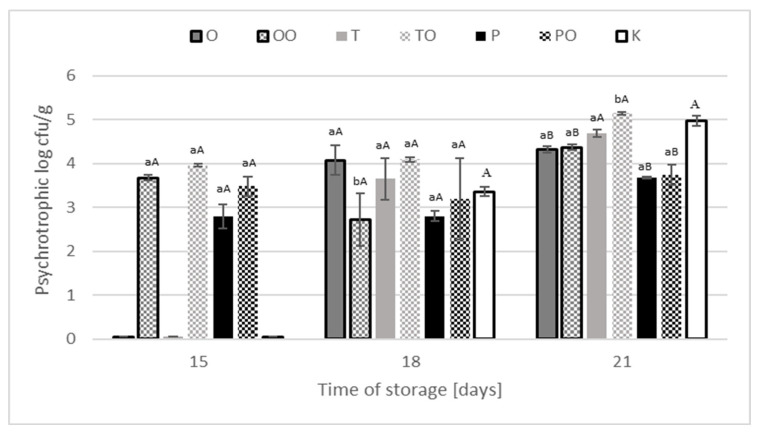
Microbial count of psychrotrophic microorganisms in treated meat products and control sample during storage at temperature 4 ± 1 °C. ^ab^ statistic differences (*p* < 0.05) in bacteria count between products with dray and oleoresin spices; ^AB^ statistic differences (*p* < 0.05) in bacteria count between products with spices and control sample. K—sample control, O—oregano, P—black pepper, T—thyme, OO—oleoresin oregano, OP—oleoresin black pepper, OT—oleoresin thyme.

**Figure 4 foods-10-01537-f004:**
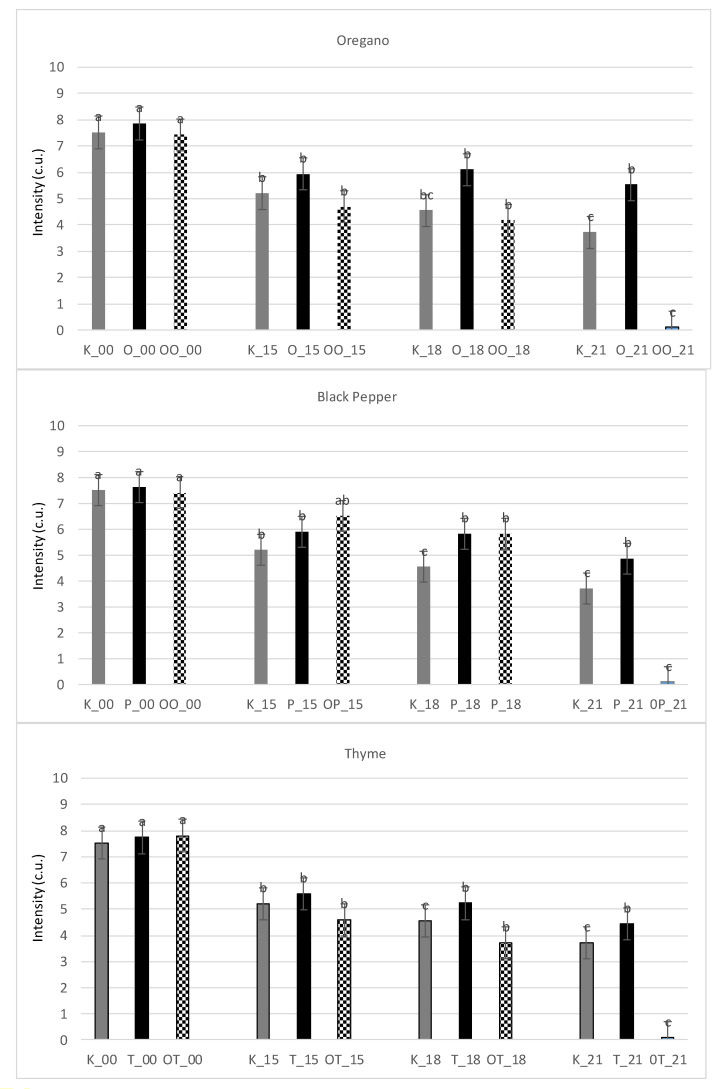
Comparison of overall sensory quality obtained in QDP method of meat products with herbal additives and industrial extracts and control during storage at temperature 4 ± 1 °C. K—sample control, O—oregano, P—black pepper, T—thyme, OO—oleoresin oregano, OP—oleoresin black pepper, OT—oleoresin thyme; 00; 15; 18; 21-storage. a, b, c—means with different superscripts differ significantly between products at various days of storage (*p* < 0.05).

**Table 1 foods-10-01537-t001:** The raw material composition of the research material and their coding (replicated twice).

Material [%]	Sample Control	Herbal Additives—Dry Spices			Industrial Extracts (Oleoresins)		
	Control	Oregano	Black Pepper	Thyme	Oregano	Black Pepper	Thyme
	K	O	P	T	OO	OP	OT
Poultry meat	80	80	80	80	80	80	80
Pork jowl	20	20	20	20	20	20	20
Salt *	1.3	1.3	1.3	1.3	1.3	1.3	1.3
Herbal/Extract addition *	0.00	0.30	0.30	0.30	0.03	0.03	0.03

K—sample control, O—oregano, P—black pepper, T—thyme, OO—oleoresin oregano, OP—oleoresin black pepper, OT—oleoresin thyme; *—in relation to meat raw material.

**Table 2 foods-10-01537-t002:** Sensory descriptors and their definitions used in the analysis of meatballs quality.

Attribute	Definition		Anchors
Odour attributes			
Heated meat	Aroma associated with heated meat	None to very strong	
Sour	Basic aroma sensation stimulated by acids	None to very strong	
Fatty	Aroma sensation derived from fat	None to very strong	
Herbal	Aroma sensation derived from herbs	None to very strong	
Other	Enter the intensity and name or association	None to very strong	
**Appearance and texture attributes**			
Surface colour	Colour impression	Light to beige dark	
Homogeneity	Impression of uniform colour	None to very strong	
Cross section colour	Colour of cross section impression	Light to beige dark	
Juiciness	Impression of moisture release during chewing	Not juicy to very juicy	
Softness	Impression of ease of separation during chewing	Not easy to very easyt	
**Flavour attributes**			
Heated meat	Flavour associated with heated meat	None to very strong	
Sour	The basic taste sensation stimulated by acids	None to very strong	
Fatty	Sensation derived from fat	None to very strong	
Salty	The basic taste sensation stimulated by salts	None to very strong	
Bitter	The basic taste sensation stimulated by capsaicin	None to very strong	
Herbal	Sensation derived from herbs	None to very strong	
Other	Enter the intensity and name or association	None to very strong	
Overall quality	Impression based on all tested attributes	Very low to very high	

**Table 3 foods-10-01537-t003:** Results of pH value (expressed as mean ± STD) in the tested material during storage time (n = 3) at temperature 4 ± 1 °C.

Storage Time (Days)	Samples							
	K	T	P	O	K	OT	OP	OO
0	6.36 ^dA^ ±0.02	6.26 ^aA^ ±0.01	6.24 ^aA^ ±0.01	6.25 ^abA^ ±0.01	6.15 ^abAB^ ±0.03	6.08 ^abB^ ±0.01	6.12 ^abcAB^ ±0.01	5.96 ^aB^ ±0.03
5	6.18 ^aA^ ±0.02	6.19 ^bA^ ±0.02	6.21 ^bA^ ±0.01	6.17 ^cA^ ±0.01	6.17 ^abA^ ±0.01	6.14 ^cAB^ ±0.01	6.13 ^bcdAB^ ±0.00	6.14 ^bAB^ ±0.01
10	6.26 ^bA^ ±0.01	6.26 ^aA^ ±0.01	6.22 ^abA^ ±0.01	6.24 ^abA^ ±0.01	6.15 ^abAB^ ±0.01	6.10 ^bB^ ±0.00	6.09 ^aB^ ±0.00	6.12 ^bB^ ±0.00
15	6.29 ^bcA^ ±0.00	6.25 ^acA^ ±0.01	6.24 ^aA^ ±0.01	6.27 ^bA^ ±0.01	6.11 ^aB^ ±0.01	6.09 ^abB^ ±0.01	6.10 ^abA^ ±0.01	6.13 ^bAB^ ±0.01
18	6.30 ^cA^ ±0.01	6.23 ^cA^ ±0.01	6.22 ^abA^ ±0.01	6.22 ^aA^ ±0.02	6.18 ^abAB^ ±0.01	6.16 ^dAB^ ±0.01	6.14 ^cdAB^ ±0.01	6.15 ^bAB^ ±0.01
21	6.21 ^aA^ ±0.01	6.20 ^bA^ ±0.01	6.16 ^cA^ ±0.01	6.23 ^aA^ ±0.01	6.18 ^abA^ ±0.01	6.08 ^aB^ ±0.02	6.15 ^dAB^ ±0.02	6.11 ^bB^ ±0.01

K—sample control, O—oregano, P—black pepper, T—thyme, OO—oleoresin oregano, OP—oleoresin black pepper, OT—oleoresin thyme; ^a, b, c, d^—means with different superscripts differ significantly in columns (*p* < 0.05). ^A, B^—means with different superscripts differ significantly in rows (*p* < 0.05).

**Table 4 foods-10-01537-t004:** Results of water activity (aw) (expressed as mean ± STD) in the tested material during storage time (n = 3) at temperature 4 ± 1 °C.

Storage Time (Days)	Samples							
	K	O	T	P	K	OO	OT	OP
0	0.886 ±0.010	0.871 ±0.020	0.877 ±0.010	0.881 ±0.017	0.925 ±0.037	0.899 ±0.040	0.911 ±0.010	0.920 ±0.012
21	0.902 ±0.025	0.873 ±0.030	0.880 ±0.010	0.894 ±0.015	0.940 ±0.050	0.909 ±0.032	0.939 ±0.011	0.946 ±0.010

K—sample control, O—oregano, P—black pepper, T—thyme, OO—oleoresin oregano, OP—oleoresin black pepper, OT—oleoresin thyme.

**Table 5 foods-10-01537-t005:** *Bacillus* spp. and total anaerobe microorganisms contamination in treated meat products and control sample (expressed as mean ± STD) during storage at temperature 4 ± 1 °C.

Storage Time (Days)	*Bacillus* spp. [log cfu/g]							Total Anaerobe Microorganisms [log cfu/g]			
	K	O	OO	T	OT	P	OP	K	OO	OT	OP
0	-	-	-	-	-	-	-	2.48 ^A^ ±0.12	0.80 ^A^ ±1.13	3.06 ^A^ ±0.21	1.48 ^A^ ±0.67
5	-	-	-	-	-	-	-	1.45 ^A^ ±0.21	1.72 ^A^ ±0.17	2.82 ^B^ ±0.15	1.72 ^A^ ±0.17
10	1.54 ^A^ ±0.13	-	-	-	0.50 * ^aA^ ±0.20	-	-	2.72 ^A^ ±0.13	1.75 ^A^ ±0.21	2.61 ^A^ ±0.06	1.63 ^B^ ±0.46
15	1.64 ^A^ ±0.18	-	3.02 ^aB^ ±0.18	-	2.70 * ^aB^ ±0.43	-	-	2.66 ^A^ ±0.32	1.96 ^A^ ±0.16	3.05 ^A^ ±0.15	2.02 ^A^ ±0.17
18	2.17 ^A^±	1.34 ^aB^±	3.29 ^bB^±	3.36 ^aB^±	1.72 ^bA^±	1.15 ^aB^±	1.42 ^aB^±	2.89 ^A^ ±0.02	0.50 ^B^ ±0.70	2.16 ^A^ ±0.14	1.72 ^A^ ±0.33
21	2.43 ^A^ ±0.01	2.11 ^aA^ ±0.40	2.61 ^aA^ ±0.23	u.	2.00 ^aA^ ±0.05	1.98 ^aA^ ±0.09	2.02 ^aA^ ±0.16	-	-	2.66 ^A^ ±0.26	-

K—sample control, O—oregano, P—black pepper, T—thyme, OO—oleoresin oregano, OT—oleoresin thyme OP—oleoresin black pepper, “-” not determinate. *—*B. cereus*; u.—microorganism uncountable; ^ab^ statistical differences (*p* < 0.05) in bacteria count between products with dry and oleoresin spices; ^AB^ statistical differences (*p* < 0.05) in bacteria count between products with spices and control sample.

**Table 6 foods-10-01537-t006:** The factor loadings of individual main components separated from MFA (multiple factor analysis) for attributes obtained in the QDP method (quantitative descriptive profile) for samples with herbal additives.

Traits	Principal Components			
	Herbal Additives		Industrial Extracts	
	1 *	2 **	1	2
meaty o.	**0.89 *^a^***	0.27	**0.94**	0.13
sour o.	**−0.93**	0.21	**−0.90**	0.15
fatty o.	−0.55	0.68	**−0.89**	−0.06
herbal o	0.00	**−0.91**	0.08	**−0.78**
other o.	**−0.90**	0.06	**−0.96**	0.02
surface c.	**−0.86**	−0.21	−0.66	−0.44
cross-section c.	0.67	0.36	−0.30	0.27
homogeneity c.	0.02	**0.92**	−0.00	**0.72**
juiciness	0.19	−0.06	−0.18	0.57
softness	0.59	−0.15	**−0.84**	0.02
meaty f.	**0.75**	0.33	**0.88**	0.27
sour f.	−0.67	−0.17	**−0.85**	0.02
fatty f.	**0.74**	−0.01	**−0.93**	0.23
salty t.	**0.76**	−0.23	−0.63	0.26
bitter t.	**−0.82**	−0.40	**0.72**	−0.03
herbal f.	0.11	**−0.97**	−0.19	**−0.80**
other f.	**−0.83**	0.10	−0.55	0.12
overall quality	**0.86**	−0.32	**0.94**	0.05
Eigen value	**8.57**	**3.85**	**9.18**	**2.62**
% of variance	**48**	**21**	**51**	**15**

O—odour, f—flavour, t.—taste, c—colour, *—first component, **—second component. *a*—values in bold (above 0.70) indicate the importance of value.

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
