# Peer review of "Effect of Herbal Addition on the Microbiological, Oxidative Stability and Sensory Quality of Minced Poultry Meat"

_foods, 2021, doi:10.3390/foods10071537_

Round 1
Reviewer 1 Report
The overall article and the subject matter are of great interest and need in the meat processing industry. The choices made for herbs and oleoresins are very good and have much merit. The parameters measured are very appropriate and of benefit to potentially replaced synthetic materials. However, the overall clarity, the statistical design and results presented are not acceptable as completed and designed. Sentence clarity, subject verb agreement, and wordiness need a lot of work throughout the entire paper. Secondly and most importantly, I saw no indication, description or analysis that would indicate multiple replications of the experiment. Some measures were mentioned in triplicate, but there should be at least three different replications of the entire experiment on different dates etc. to account for variability due to day. Without multiple replications of all treatments, you cannot make any sound assumptions on the analysis. Also, the analysis should be made across treatments within each day of measurement and across treatments within day to adequately compare each treatment to each other. All this should be fully and clearly explained for each different parameter analysis. I have made numerous corrections, suggestions, edits and comments within the text of the paper and have attached my edited copy of the article. I hope you consider these comments constructively and incorporate them into the rewriting and reworking of the experiment since it is a topic of good merit. Please consult with statistician about proper placement of significance letters and interpretation as some do not seem to make logical sense, depending on how the analysis was done... which was not clear in the manuscript; within treatment across times or across treatments within storage times. There needs to be more detail included in the materials and methods about how the product was obtained, processed, formed into meatballs, packaged for storing, where and how held during storage time (under lights, in dark, etc. etc.) and how prepared in detail for sensory analysis. Randomization of presenting samples to panelists, how many samples per evaluation etc. etc... There needs to be much more discussion and writing about the results depending on what is significanlty different about the attributes. The tables need more subscripts describing what is in the tables, figures etc. especially about the significance levels etc. Some listings as graphs are confusing with the identification of the treatments and should be listed as numbers and or spread out more for clarity and ease of understanding. Discussion about significance or not should be very clear with P values in tables.... Be sure that table and figure headings actually depict and tell what is in the table or figure and not something else that is not shown or represented. This is evident in figures 4 and 5. Graphs with the four quadrants showing the sensory parameters may be a better depiction for the attribute analysis. Be careful with broad statements in the conclusions as the data only represent a few spices in a specific application.

Author Response
Open Review
(x) I would not like to sign my review report
( ) I would like to sign my review report
English language and style
( ) Extensive editing of English language and style required
(x) Moderate English changes required
( ) English language and style are fine/minor spell check required
( ) I don't feel qualified to judge about the English language and style
|
Yes |
Can be improved |
Must be improved |
Not applicable |
|
|
Does the introduction provide sufficient background and include all relevant references? |
(x) |
( ) |
( ) |
( ) |
|
Is the research design appropriate? |
( ) |
(x) |
( ) |
( ) |
|
Are the methods adequately described? |
( ) |
(x) |
( ) |
( ) |
|
Are the results clearly presented? |
( ) |
(x) |
( ) |
( ) |
|
Are the conclusions supported by the results? |
( ) |
(x) |
( ) |
( ) |
Comments and Suggestions for Authors
Major comments
These are the interesting results, but somewhat lacking in impact. From these data I suppose it is difficult to greatly extend the expiration date by herbal addition.
Dear Reviewer
Thank you very much for your thorough review of our paper. Thank you for your patience and attention. Your detailed comments enabled us to improve our manuscript. The responses to your comments are listed below, all the changes are included in the new version of our manuscript
Response to comments from reviewers:
The comments and responses to Reviewers` comments are listed below. All the changes are included in the new version of our manuscript and are marked in red.
What is the composition of the oleoresin? Does it contain oil, antioxidant, and taste/aroma components? Is there possibility that the effect is due to what is added industrially?
Oleoresin are composed of essential oils dissolved in oils and based on the manufacturer's specifications oregano contains 45 – 65ml/100 g essential oils, black pepper - 40 – 41 % of piperine and min. 25 ml/100 g essential oils and thyme contains 40-50% essential oils. All specimen are declared free from antioxidants addition.
About the experimental materials (meatballs), many contaminations were seen during storage. Were the manufacturing and storage methods appropriate?
The manufacturing method, raw material quality as well as storage methods were unreservedly in our opinion. However the contaminations concerned dry herbs what is common and described in the literature (Ergün and Baysal. 2017; Krisnan, et al. 2014).
How about combining Figures 4 and 5 into a single graph? Readers want to compare the effects of herbal additives and industrial extract. Isn't it subjective that the dry herbal additives are more effective in quality preservation compared to industrial extracts. Is it statistically different?
Thank you for your suggestion. We have prepared according your advice a new Figure 5 which will be more suitable for this experiment. Sometimes too many results make it difficult to present them clearly. In order to make the Figure readable, we limited the number of presented data. We skipped the storage step up to 10 days when no changes were observed.
In the description of stastical methods, there is little information about MFA.
We have improved the approach in interpreting the results of sensory research.
I don't understand the need for Table 5 and Table 6. Readers want to know the results of how adding herbs changes the taste and aroma. Please describe the results in an easy-to-understand manner with or without Table.
The aim of using multidimensional statistical analysis MFA was to assess the contribution and strength of the impact of individual attributes in creating and explaining the general variability of the sensory quality of the studied material. In our opinion such a results are of important value. They explain which traits have a most important effect on overall sensory quality. They show that aroma and flavor are crucial. After careful consideration, we decided to improve Tables 5 and 6 as they are extensive. We prepare a new Table which presenting only 2 first most important principal components, which present most significant attributes.
Minor comments
A description of the results in sensory evaluation should also be included in the abstract.
We added a new sentence regarding sensory evaluation.
Please describe about the letter “a” and “b” in the Table 4. Is the footnote incorrect?
All Tables and Figures are corrected and footnotes are completed.
Please describe about the letter “c” and Eigen value in the Table 5 and 6.
In the Table 5, % of variance is 48, 21, 8, and 7, isn’t it?
Thank for the good point; we improved the Tables.
Submission Date
21 April 2021
Date of this review
03 May 2021 19:16:15

Reviewer 2 Report
See attached file.

Author Response
Open Review
(x) I would not like to sign my review report
( ) I would like to sign my review report
English language and style
( ) Extensive editing of English language and style required
(x) Moderate English changes required
( ) English language and style are fine/minor spell check required
( ) I don't feel qualified to judge about the English language and style
|
Yes |
Can be improved |
Must be improved |
Not applicable |
|
|
Does the introduction provide sufficient background and include all relevant references? |
( ) |
(x) |
( ) |
( ) |
|
Is the research design appropriate? |
( ) |
( ) |
(x) |
( ) |
|
Are the methods adequately described? |
( ) |
(x) |
( ) |
( ) |
|
Are the results clearly presented? |
( ) |
( ) |
(x) |
( ) |
|
Are the conclusions supported by the results? |
( ) |
( ) |
(x) |
( ) |
Comments and Suggestions for Authors
The objective of the study is to investigate the preservative effect of different herbs on minced poultry meat. Several antimicrobial systems were tested (i.e. oregano, black-pepper, thyme) as dry spices and as oleoresins.
The study is interesting and well designed. However, the authors should consider the following issues and revise the manuscript.
Dear Reviewer
Thank you very much for your thorough review of our paper. Thank you for your patience and attention. Your detailed comments enabled us to improve our manuscript. The responses to your comments are listed below, all the changes are included in the new version of our manuscript
Response to comments from reviewers:
The comments and responses to Reviewers` comments are listed below. All the changes are included in the new version of our manuscript and are marked in red.
Based on the obtained results, it is evident that the two Controls deviate significantly between them. Since a batch deviation is observed (the standard deviations should be also reported in all presented data in Tables and Figures), how can someone conclude in the statistical analysis shown in Tables 2 and 3?
According to our long experience in research and laboratory practice the only the model material is of repeatable quality. Likewise, many years of experience with raw meat shows that this product is characterized by natural variability. Due to the duration of this experiment (almost 2 years), the variability of the results obtained in the control trials may be justified by the variability of the raw material used for the production of the tested meatballs.
We completed standard deviation in presented data what was overlooked.
Please include the error bars in Figures 1, 2 and 3.
According your advice we included the error bars in mentioned Figures.
Statistics should be also included in the data presented in the Tables.
All Tables and Figures are improved or simply changed or limited.
Please indicate the storage temperature in the Figure and Table legends.
We added the required data.
English language should be carefully revised throughout the manuscript.
You have to wait for a specialist linguistic proofreading. We had a very limited time to improve our work. Given the large amount of work to improve the manuscript; we have to prepared almost all new tables, there was no time to send the paper a linguistic professional correction but we did our best. We improved the language. We ask for your understanding.
Submission Date
21 April 2021
Date of this review
29 Apr 2021 09:06:32

Reviewer 3 Report
Major comments
These are the interesting results, but somewhat lacking in impact. From these data I suppose it is difficult to greatly extend the expiration date by herbal addition.
What is the composition of the oleoresin? Does it contain oil, antioxidant, and taste/aroma components? Is there possibility that the effect is due to what is added industrially?
About the experimental materials (meatballs), many contaminations were seen during storage. Were the manufacturing and storage methods appropriate?
How about combining Figures 4 and 5 into a single graph? Readers want to compare the effects of herbal additives and industrial extract. Isn't it subjective that the dry herbal additives are more effective in quality preservation compared to industrial extracts. Is it statistically different?
In the description of stastical methods, there is little information about MFA.
I don't understand the need for Table 5 and Table 6. Readers want to know the results of how adding herbs changes the taste and aroma. Please describe the results in an easy-to-understand manner with or without Table.
Minor comments
A description of the results in sensory evaluation should also be included in the abstract.
Please describe about the letter “a” and “b” in the Table 4. Is the footnote incorrect?
Please describe about the letter “c” and Eigen value in the Table 5 and 6.
In the Table 5, % of variance is 48, 21, 8, and 7, isn’t it?
Author Response
Open Review
( ) I would not like to sign my review report
(x) I would like to sign my review report
English language and style
( ) Extensive editing of English language and style required
( ) Moderate English changes required
( ) English language and style are fine/minor spell check required
(x) I don't feel qualified to judge about the English language and style
|
Yes |
Can be improved |
Must be improved |
Not applicable |
|
|
Does the introduction provide sufficient background and include all relevant references? |
(x) |
( ) |
( ) |
( ) |
|
Is the research design appropriate? |
(x) |
( ) |
( ) |
( ) |
|
Are the methods adequately described? |
(x) |
( ) |
( ) |
( ) |
|
Are the results clearly presented? |
( ) |
(x) |
( ) |
( ) |
|
Are the conclusions supported by the results? |
(x) |
( ) |
( ) |
( ) |
Comments and Suggestions for Authors
Dear authors, I do not have any major concerns regarding this manuscript, there are just some minor technical details on which you need to pay attention. Here are they:
Dear Reviewer
Thank you very much for your thorough review of our paper. Thank you for your patience and attention. Your detailed comments enabled us to improve our manuscript. The responses to your comments are listed below, all the changes are included in the new version of our manuscript
Response to comments from reviewers:
The comments and responses to Reviewers` comments are listed below. All the changes are included in the new version of our manuscript and are marked in red.
Page 6. Table 2 has superscripts - there is no explanation on how to read them, please add an explanation.
All Tables in the manuscript has been improved.
Figure 3 - "Time of dtorage" should be renamed
Also in all Figures we made corrections.
Line 369 - Sentence "Similarly, in study of [31]..." it would be better to insert authors than just a reference number, and you do not need the same reference at the end of the same sentence.
We have improved the indicated sentence.
Line 436 - Add reference number after the authors
The mentioned reference was added.
Line 439 - Reference Krisnan et al., 2014) - I presume this was mistakenly left during the writing process, please remove and correct the reference number, Krisnan is number 41, not 42.
This mistake was corrected.
Due to the reference mistake noticed, please also once more double-check all references and their numbers.
We have checked again the list of references and improved it.

Reviewer 4 Report
The objective of the study is to investigate the preservative effect of different herbs on minced poultry meat. Several antimicrobial systems were tested (i.e. oregano, black-pepper, thyme) as dry spices and as oleoresins.
The study is interesting and well designed. However, the authors should consider the following issues and revise the manuscript.
Based on the obtained results, it is evident that the two Controls deviate significantly between them. Since a batch deviation is observed (the standard deviations should be also reported in all presented data in Tables and Figures), how can someone conclude in the statistical analysis shown in Tables 2 and 3?
Please include the error bars in Figures 1, 2 and 3.
Statistics should be also included in the data presented in the Tables.
Please indicate the storage temperature in the Figure and Table legends.
English language should be carefully revised throughout the manuscript.
Author Response

(The authors gave the same response as above.)

Reviewer 5 Report
Dear authors, I do not have any major concerns regarding this manuscript, there are just some minor technical details on which you need to pay attention. Here are they:
Page 6. Table 2 has superscripts - there is no explanation on how to read them, please add an explanation.
Figure 3 - "Time of dtorage" should be renamed
Line 369 - Sentence "Similarly, in study of [31]..." it would be better to insert authors than just a reference number, and you do not need the same reference at the end of the same sentence.
Line 436 - Add reference number after the authors
Line 439 - Reference Krisnan et al., 2014) - I presume this was mistakenly left during the writing process, please remove and correct the reference number, Krisnan is number 41, not 42.
Due to the reference mistake noticed, please also once more double-check all references and their numbers.
Author Response

(The authors gave the same response as above.)

Round 2
Reviewer 1 Report
I HAVE DONE A BRIEF REVIEW OF THE REVISED ARTICLE REFERENCED ABOVE AND HAVE FOUND THAT THE ONLY CHANGES I HAD SUGGESTED BE MADE WERE THE SAME AS SOME OF THE OTHER REVIEWERS. IT IS OBVIOUS THAT THE AUTHORS ( 1. ) IGNORED MY NUMEROUS COMMENTS WRITTEN ON THE ACTUAL ARTICLE WHICH I SCANNED AND DOWNLOADED BACK WITH MY COMMENTS IN THE REVIEWER FORM AND MY NUMEROUS SUGGESTIONS MADE ON THE REVIEWER FORM OR ( 2. )THE SCANNED ARTICLE WHICH I SPENT A GREAT DEAL OF TIME MAKING GRAMMATICAL CHANGES AND SUGGESTIONS ALONG WITH NUMEROUS OTHER COMMENTS ON THE ACTUAL ARTICLE WAS NOT FORWARDED TO THE AUTHORS.
It is riddled with grammatical errors and subject verb disagreements. If the authors cannot follow my hand corrected version, then I suggest they find someone who is fluent in the English language writing style and seek their help in revising this manuscript. There are also still numerous questions about the format of the statistical analysis and/or the explanations as well as inconsistencies of using comas instead of decimals throughout the tables with numerical values. I am attaching the scanned version of my initial edit again for their ease and help with editing.

Author Response
Dear Reviewer,
We appreciate your whole work and your commitment to improving our manuscript.
We are very sorry not correct your all comments and suggestion in a previous version of our paper. I have never ever ignore someone work or comments. We have not get any scanned document. We only got it after we contacted the editorial office again.
We are so sorry for the really awkward situation.
We have made another effort to improve the details (statistical analysis, explanation and all inconsistencies) in our work and we hope that we have met the requirements. We add a new Table with list of attributes according your advice.
Thank you again for your help and understanding.

Reviewer 2 Report
No SEM or STD Dev for Tables 2 and 3. For Fig 1. Only Day 0 and 21 reported for TBARS. No SEM/STD DEV. Stated that all analyses were conducted on days 0, 5, 10, 15, 18 and 21 of tested product. In the discussion section the only main effect discussed was spices/herbs. No discussion of whether length of storage had any effect on lipid oxidation. Makes me question if there was a two-way interaction. Figure 5. No Day 21 results reported. Table 5 and 6 do not indicate to reader the importance of the bolded PCA values. Each table/figure should clearly communicate to the reader what the values mean. There is no descriptive legend for these. Figure 1 2 and 3 are done correctly. Still grammatical errors throughout paper. The authors’ response for my review was exactly the same for another reviewer. Although I appreciate the authors’ efforts in correcting the manuscript
Author Response
Dear Reviewer,
We have made another effort to improve all the details (statistical analysis, explanation and all inconsistencies) in our work.
We tried to improve again all Figures and Tables. We added STD Dev for tables and all information to communicate to the reader meaning of values.
On Figure 1 are data reported for TBARS only for a day 0 and 21 as no statistical changes were stated in day 5; 10; 15 and 18.
In order to make the Figures 4 readable, we limited the number of presented data. We skipped the storage step up to 10 days when no changes were observed.
On the Figure 4 the day 21 is reported. The values were “0,0” as the studied material was spoiled and unsuitable for sensory evaluation. It is mentioned in the text and marked on a Figure.
We have not conduct two-way interaction. The importance of the bolded PCA values is above 0.7. It is improved.
We improved discussion section.
We have send our paper for linguistic proofreading.
Thank you very much your opinion and help by improving this manuscript

Reviewer 4 Report
The authors addressed adequately the reviewers' suggestions. The article can be accepted for publication after a careful revision of English language.
Author Response
Dear Reviewer,
Thank you very much for your comments and help.
The manuscript was careful revised of language.
Thank you again.